# Characterization and deorphanization of RYamide signaling in *Aedes aegypti*: A potential regulator of hindgut-associated physiology

**Jinghan Tan, Thomas Luong, Jean-Paul V. Paluzzi**[ID]*

Department of Biology, York University, Toronto, Canada

* paluzzi@yorku.ca

## Abstract

Neuropeptide Y-related signaling, consisting of various neuropeptides and their receptors including, for example, the mammalian neuropeptide Y (NPY), pancreatic polypeptide (PP), and peptide YY (PYY) families, along with neuropeptide Y-like peptides in insects, is evolutionarily conserved across vertebrates and invertebrate organisms. Vertebrate NPY is known as an important regulator of energy homeostasis and feeding behaviour, while studies on one NPY-associated signaling system in arthropods, known as RYamide, have mainly focused on functions regulating feeding-related behaviours. The current study aimed to begin exploring an additional functional role of RYamide signaling related to excretory physiology in *Aedes aegypti* given that a candidate RYamide receptor, RYaR2, is enriched in the hindgut of adult mosquitoes based on publicly available RNA-seq databases. Herein, we report that the *RYamide* transcript is expressed in all post-embryonic stages with greatest abundance in adult male mosquitoes. Specifically, the central nervous system and the ventral nerve cord were demonstrated as the major sources of RYamide, as supported by RT-qPCR and intense RYamide immunoreactivity observed in the protocerebral posterior dorsomedial neurons and protocerebral anterolateral neurons in the mushroom body region, along with an intensively stained pair of neurons in the terminal abdominal ganglion (TAG). The pair of immunoreactive neurons in the TAG contain elaborate processes extending onto the hindgut, with a branch extending posteriorly over the rectum where fine processes are in close proximity to the rectal papillae (pads) while an anterior branch extends over the ileum terminating at the midgut-hindgut junction forming a fine circular plexus over the pyloric valve. Importantly, two orphan GPCRs, previously referred to as NPYLRs, were functionally deorphanized as *bona fide* RYamide receptors. Their responses to diverse peptidergic ligands were assessed, revealing that RYamides elicited the strongest receptor activation, with half maximal effective concentrations in the pico- to nanomolar range. Notably, our phylogenetic analysis revealed that while only a single RYaR is commonly found across most arthropods, culicine mosquito species including *A. aegypti*

**Data availability statement:** All relevant data are within the manuscript and its Supporting Information files.

**Funding:** This research was funded by the Natural Sciences and Engineering Research Council of Canada (NSERC) Discovery Grant (JPP), Ontario Ministry of Research Innovation Early Researcher Award (JPP). The funders had no role in study design, data collection and analysis, decision to publish, or preparation of the manuscript.

**Competing interests:** The authors have declared that no competing interests exist.

possess two RYaRs. The characterization of RYamide signaling with respect to excretory physiology involving the hindgut, particularly in species having two receptor isoforms such as culicine mosquitos, may provide valuable insights for development of novel, species-specific pest control strategies.

## 1. Introduction

The yellow fever mosquito, *Aedes aegypti*, an African originated mosquito species that is widely distributed in tropical and sub-tropical regions [1,2], is recognized as a principal vector capable of transmitting dengue, yellow fever, chikungunya and other disease agents through its blood feeding behaviour [3,4]. Expanded habitats have led to an increased risk of mosquito-borne diseases among human populations [5–9]. In addition to current mosquito control strategies, such as *Wolbachia* infection [10], male sterilization [11], and traditional insecticides [12], there is a growing need to develop novel, more targeted approaches for effective mosquito control. Recently, neuropeptide Y-like receptor 7 (NPYLR7), but not a closely related NPY-like receptor 5 (NPYLR5), has been shown to act as a key regulator of host-seeking behaviour in female mosquitoes, with its activation by a small-molecule agonist resulting in suppression of host-seeking and blood feeding [13,14], making it a potential target for pest management. Nonetheless, despite NPYLR7 being activated by a human neuropeptide Y (NPY) Y2 receptor agonist and several synthetic small-molecule agonists [13], it remained an orphan receptor in mosquitoes as no endogenous ligands were identified.

While the endogenous peptidergic ligands for NPYLR5 and NPYLR7 remain unknown in mosquitoes, earlier phylogenetic analysis examining luqin-type neuropeptide signaling provided evidence that NPYLR5 belongs to an orthologous group that includes receptors for arthropod RYamide receptors [15–17]. The RYamide prepropeptide, encoded by a single gene, is widely conserved across arthropods [18], such as *Drosophila melanogaster* [19,20], *Ae. aegypti*, *Bombyx mori* [21]. The prepropeptide in most insects, including *Ae. aegypti*, is typically cleaved forming only two peptide isoforms of RYamide and/or RFamide, which share a conserved FFXXXRY/Famide motif [18]. RYamide shares a similar C-terminal sequence (GxRYamide, where X is S in most examined species) [18] with insect neuropeptide F (NPF) peptides that often contain RxRFamide, where X is often P, A or V and terminal F sometimes substituted with a Y residue [22], which are related to the vertebrate NPY family (NPY, pancreatic polypeptide (PP), and peptide YY (PYY)) known to regulate feeding behaviour, energy expenditure and energy homeostasis [23–25]. Additionally, other neuropeptides containing an RF or RYamide motif at the C-terminus, such as short neuropeptide F (sNPF) [26,27], have also been identified in mosquitoes. However, there is no evidence suggesting that the genes encoding RYamide, sNPF, and NPF share a common evolutionary origin [18,22,28].

The localization of RYamide demonstrated that the brain is an important source of RYamide production in insects and RNA-seq data have confirmed a high *RYamide* transcript (AAEL011702) abundance in the adult mosquito brain [21,29,30]. In

addition to its distribution within the brain, two RYamide producing neurons have been immunolocalized in the abdominal ganglia of adult *D. melanogaster*, with projections extending towards the hindgut [20], an organ responsible for modifying primary urine through selective reabsorption of nutrients, ions and water [31–36]. In insects, innervation of the hindgut from the terminal abdominal ganglion has been reported [37,38], suggesting that RYamide may play a role in regulating excretory physiology and hydromineral homeostasis. Corroborating this hypothesis, RYaR (CG5811) transcript is highly enriched in the hindgut, including in the rectal pads [39,40], which are known for intensive ion transport activities within the hindgut [41–45]. It is worth noting that the second RYamide receptor (AAEL019786; NPYLR7) candidate examined in the current study, which was previously reported to respond to sNPF and human PYY at relatively high concentrations [46], is highly abundant in the adult mosquito hindgut [29]. However, previous studies in insects have predominantly focused on the role of RYamides in suppressing feeding-related behaviours [13,14,19]. Additionally, electrophysiological recordings revealed a significant reduction in sugar receptor neuron activity in flies following RYamide knockdown. [47]. Aside from feeding behaviour-related actions, RYamides appear to be involved in regulating pharynx and ileum muscle contractions that potentially relate to feeding and digestion in fed silkworm larvae [48].

To explore potential osmoregulatory role of RYamide signaling and to gain a better understanding of excretory physiology in mosquitoes, we investigated transcript expression of RYamide over the course of mosquito development and in various tissues of adult males and females including the nervous system and peripheral organs. Using a custom antibody generated against *Ae. aegypti* RYamides, we mapped immunoreactivity within neurons in the nervous system including the brain and terminal abdominal ganglia as well as axonal processes associated with excretory organs including the hindgut. Importantly, our study set out to deorphanize two *Ae. aegypti* RYamide receptor candidates by assessing the specificity and ligand activity of endogenous RYamides as well as several structurally related neuropeptides. This is particularly important for the RYamides since they share structural similarity with a number of distinct insect neuropeptide families, including FMRFamides (FMRFa) [49,50], neuropeptide F (NPF) [51,52], short neuropeptide F (sNPF) [22,27], sulfakinins [53,54], and myosuppressins [55,56]. Lastly, the current study aims to understand the phylogeny and evolutionary history of RYamide receptors across arthropods (and more broadly in other animals), which was prompted by the presence of two RYamide receptors in *Ae. aegypti*, whereas all other reported insect species to date, including *D. melanogaster*, *T. castaneum*, and *B. mori*, possess only a single receptor [39,57].

## 2. Materials and methods

### 2.1 Animal rearing

A colony of *Ae. aegypti* (Liverpool strain) was maintained by regularly providing female mosquitoes with sheep's blood in Alsever's solution (Cedarlane Laboratories Ltd., Burlington, ON), which allows them to produce and later deposit eggs on dampened filter papers (Fisher Scientific) placed within BugDorm insect cages (MegaView Science Co., Ltd). Eggs were collected and hatched in plastic containers with dechlorinated water, and reared in an incubator at 25°C on a 12:12hr light: dark cycle as previously described [58]. Larval food was prepared containing 2% (w/v) brewer's yeast (Bulk Barn, Milton, ON) and 2% (w/v) beef liver powder (NOW Foods, Bloomingdale, Illinois) suspended in distilled water with ~2mL of this suspension provided daily to each litre of larval bath. Adult mosquitoes were provided 10% sucrose solution in a small petri-dish covered with parafilm and fitted with a cotton ball allowing them to feed *ad libitum*.

### 2.2 Design of primers for qPCR, receptor expression constructs and dsRNA synthesis

Primers for reverse transcriptase quantitative PCR (RT-qPCR), synthesizing double-strand RNA (dsRNA) that targets the *RYamide* transcript, and mammalian expression constructs containing the two candidate RYamide receptors, specifically AAEL017005, NPYLR5 and AAEL019786, NPYLR7 (see Supplementary S1 Table) were designed using the Primer3 tool in Geneious Pro Bioinformatics Software (Biomatters Ltd). The amplification efficiency of RT-qPCR primer pairs was

determined by a standard curve analysis using head cDNA from adult female mosquitoes. Specificity of amplification was confirmed by observing no detection in negative controls including a no reverse transcriptase control group that accounts for false positive amplification from potential genomic DNA contamination and a no template control that eliminates the possibility of any non-specific amplification including primer-dimers.

## 2.3 Profiling developmental and spatial expression of RYamides via RT-quantitative PCR

Whole body of *Ae. aegypti* of 4th larva, early and late pupa, one- and four-day-old adults (10 individuals per biological replicate), and four-day old dsRNA-injected adults (72h post-injection) were collected and stored in 1X RNA protection buffer for subsequent RNA extraction (Monarch Total RNA Miniprep kit, New England Biolabs, Whitby, ON). Tissues/organs (head, thorax, midgut, Malpighian tubules (MTs), reproductive system including primary and accessory reproductive organs, and remaining carcass including the cuticle with underlying epidermis, skeletal muscle, fat body and abdominal ganglia of anaesthetized one-day-old adult mosquitoes (40 individuals per biological replicate) were dissected in Dulbecco's phosphate buffered saline (DPBS; Wisent Corporation, St. Bruno, QC, Canada). Total RNA of whole animal body and dissected tissues/organs was extracted using a Monarch® Total RNA Miniprep Kit (New England Biolabs, Whitby, ON, Canada) following the manufacture's protocol. The RNA concentration was quantified using a Synergy 2 Multi-Mode Microplate Reader (BioTek, Winooski, VT, USA) and cDNA was synthesized from purified total RNA using 500ng for whole insect samples and 15–50ng for tissue/organ samples, using iScript™ Reverse Transcription Supermix (Bio-Rad, Mississauga, ON). Synthesized cDNA was then diluted at least 10-fold with nuclease-free water prior to RT-qPCR analysis.

Transcript abundance of *RYamide* (AAEL011702) was assessed on a StepOnePlus Real-Time PCR System (Applied Biosystems, Carlsbad, CA, United States) using PowerUP™ SYBR® Green Master Mix (Applied Biosystems, Carlsbad, CA, United States). RT-qPCR cycling condition were as follow: 95°C for 20 seconds, and then 40 cycles of (i) 95°C for 3 seconds and (ii) 62°C for 30 seconds. Through ΔΔCT method the expression of RYamide was normalized using two reference genes, ribosomal protein 49 (rp49) and rpS18, as optimal endogenous reference genes [59]. To confirm reproducibility of the results, 9 biological replicates were collected for determining developmental transcript abundance of RYamides, 4–5 biological replicates for spatial expression profiles, and 5 biological replicates for validating RYamide knockdown efficiency.

## 2.4 Whole mount immunohistochemistry

To examine cellular distribution of RYamide in adult *Ae. aegypti*, the brain, gut, and the abdominal ganglia were dissected in nuclease-free Dulbecco's phosphate-buffered saline (DPBS; Wisent Corporation, St. Bruno, QC, Canada) on a silicone-lined petri dish and then fixed in 4% paraformaldehyde (PFA) overnight at 4°C. Following overnight fixation, tissues were washed three times for 10 minutes each with DPBS followed by a 1h incubation with 4% Triton X-100, 2% bovine serum albumin (BSA) and 10% normal sheep serum (NSS) on a rocking platform at room temperature. Permeabilized tissues were incubated with primary antibody anti-*Aedae*RYa1 for 48h at 4°C on a rocking platform with gentle agitation. The custom rabbit polyclonal primary antibody, anti-*Aedae*RYa1 (Biomatik Corp., Kitchener, ON) (5 µg/ml), was prepared 12h prior to use in a solution containing 0.4% Triton X-100, 2% bovine serum albumin (BSA), and 10% normal sheep serum (NSS) with 10µM synthetic *Aedes* neuropeptides, including NPF, sNPF, FMRFa3, CAPA and pyrokinin1, to reduce the possibility of cross-reactivity in immunolocalization (see Supplementary S2 Table). Following primary antibody incubation, DPBS washed tissues (three times) were treated with a goat anti-rabbit Alexa Fluor® 568 IgG (H+L) conjugated secondary antibody (Molecular Probes, Life Technologies, Eugene, OR, USA) diluted 1:200 in 10% NSS and DPBS, incubating for another 48h at 4°C on a rocker. After three times DPBS rinses (10 minutes each), tissues were mounted onto microscope slides using a mounting buffer containing 50% glycerol in DPBS and 4′,6-diamidino-2-phenylindole dihydrochloride (DAPI) (Molecular Probes, Eugene, OR, USA) to stain cell nuclei. A Lumen Dynamics XCite™ 120Q Nikon fluorescence microscope (Nikon, Mississauga, ON, Canada) and a Zeiss Laser Scanning LSM 700 confocal microscope (Carl Zeiss Canada Ltd., Toronto, ON, Canada) were used to examine samples and acquire images.

## 2.5 Preparation of RYamide receptor mammalian expression constructs

The full open reading frame of two putative *Ae. aegypti* RYamide receptors (AAEL017005; NPYLR5 and AAEL019786; NPYLR7) were amplified using high-fidelity Q5 polymerase (New England Biolabs, Whitby, ON) as described previously [60] and primers incorporating a Kozak translation initiation sequence [61], stop codon and restriction sites for directional subcloning (see Supplementary S1 Table). The digested amplicons were then subcloned into mammalian expression vector pcDNA3.1+, followed by bacterial transformation and plasmid DNA extraction to obtain a large quantity of receptor constructs using ZymoPURE II Plasmid Midiprep Kit (Zymo Research, Tustin, CA, USA) [62]. Prepared constructs were sequenced to verify base accuracy by whole plasmid sequencing performed by Plasmidsaurus using Oxford Nanopore Technology with custom analysis and annotation.

## 2.6 Heterologous functional activity bioluminescence assay

At 48h prior to the bioluminescence assay, a previously established Chinese hamster ovary cell line K1 (CHO-K1) stably expressing the calcium bioluminescent reporter protein aequorin [63] was seeded into T75 culture flasks at 90% confluency, and followed by a 6h incubation at 37°C. Following the manufacture's recommendation, mammalian expression constructs were then prepared for transfection using PolyJet™ DNA *in vitro* transfection reagent (Frogga-Bio Inc., Concord, ON). Cells were prepared for the functional activity bioluminescence assay 42h post-transfection by incubating in assay media (BSA media) containing DMEM: F12 basal media, 10% Bovine Serum Albumin (BSA) and 1X antibiotic-antimycotic, with 5µM coelenterazine-h, in the dark at room temperature for 3 hours. After a 10-fold dilution in BSA media, cells were incubated an additional 45 minutes in the dark. A library of mosquito neuropeptides (Supplementary S2 Table) were commercially synthesized (Shanghai Royobiotech Co., Ltd, Shanghai, China) at purity >95% and used initially at a final titre of $10^{-6}$M to examine the ligand selectivity and responsiveness of the two putative RYaRs. Dose-dependent responses of neuropeptides were then investigated by measuring bioluminescent responses of serial dilutions ranging from $10^{-5}$M to $10^{-16}$M. As a positive control, 100µM ATP was used to normalize the bioluminescent response in individual wells triggered by endogenous purinoceptors expressed in CHO-K1 cells [64]. Measurements of $Ca^{2+}$ kinetics was acquired with a Synergy 2 Multi-Mode Microplate Reader (BioTek, Winooski, VT, USA).

## 2.7 RYamide receptor phylogenetic analysis

RYamide receptor sequences and tachykinin receptor sequences from various species across arthropods were collected on NCBI website (https://www.ncbi.nlm.nih.gov/) by blast search of deduced *Ae. aegypti* RYaR1 (AAEL017005) and RYaR2 (AAEL019786) protein sequences and *D. melanogaster* tachykinin-like receptor (AAF56979.2) within the database. Following the protocol described previously [58], protein sequences were aligned using ClustalX2 [65]. Evolutionary relationships between receptor sequences among examined species were explored by maximum-likelihood phylogenetic analysis methods supported by 1000 bootstraps using MEGA-X software [66,67].

## 2.8 Statistical analysis

One-way ANOVA with Tukey's multiple comparisons test or two-way ANOVA with Šídák's multiple comparisons test was used to analyze the log-transformed developmental/tissue-specific transcript expression data, and luminescent responses from the functional activity bioluminescence assay, respectively. A two-tailed t-test was used to assess significance of *RYamide* knockdown as determined by RT-qPCR analysis. Dose-response curves and $EC_{50}$ values were determined using the sigmoidal dose-response model as previously described [60]. Statistical analysis was conducted using Graph-Pad Prism 9 adopting a 0.05 significance threshold.

## 3. Results

### 3.1 Expression profiles of *Ae. aegypti* RYamide over developmental stages and in adult organs

The *RYamide* transcript was expressed throughout the post-embryonic development of *Ae. aegypti*, with a relatively consistent transcript abundance. However, males at both early (1-day old) and later (4-day old) adult stages showed significantly higher transcript abundance compared to the 4$^{th}$ instar larvae (Fig 1). In addition, the *RYamide* transcript showed differential expression between adult males and females, with a significant difference between sexes in 4-day old adults.

Organ-specific transcript expression profiles of RYamide in one-day old mosquitoes was examined, revealing consistent abundance profile between both sexes. The nervous system appeared to be a primary source of *RYamide* transcript production, as the head (brain) showed high enrichment of the transcript (Fig 2). Additionally, moderate *RYamide* transcript abundance was detected in the thorax and carcass, that contain the thoracic and abdominal ganglia (respectively) as well as other tissues including (but not limited to) fat body and epidermis. In comparison with the *RYamide* transcript enrichment in the head, a significantly lower abundance was detected in the hindgut and reproductive tissues of male mosquitoes (Fig 2A). A similar trend was noted in female mosquitoes, although without any statistical significance (Fig 2B). Across all tested tissues and in both sexes, MTs had the lowest *RYamide* transcript abundance detected with significantly lower levels compared to the brain. This expression profile aligns with publicly available RNA-seq data, which indicates enrichment in the head of female *Ae. aegypti* (Supplementary S1 Fig).

### 3.2 Deorphanization of RYamide receptors

The prepared mammalian expression constructs were validated by comparing the responsiveness of cells transfected with the control construct, pBud-EGFP, compared to cells transfected with each of the RYamide receptor constructs. The RYaR1 and RYaR2 transfected cells exhibited strong luminescent response following application of RYamides while no luminescent response to RYamides was observed in cells transfected with the control construct, pBud-EGFP (Supplementary S2 Fig).

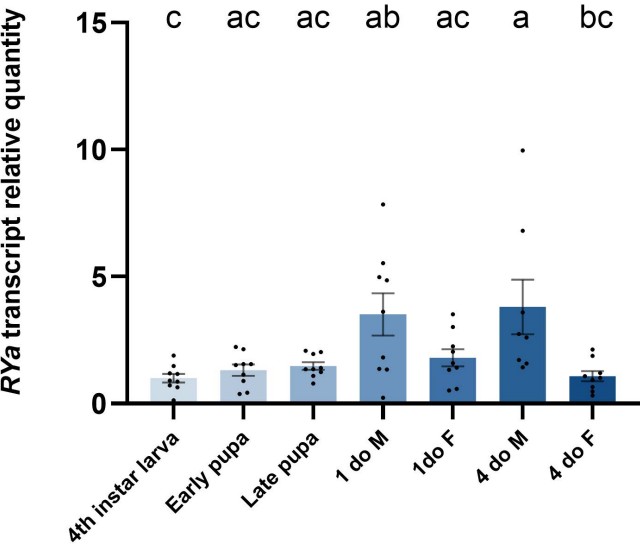

**Fig 1. Expression profile of *Ae. aegypti* RYamide (RYa) transcript in different post-embryonic developmental stages determined by RT-qPCR.** Transcript levels of various developmental stages relative to the fourth instar larva. Abbreviations: 1do M = 1-day-old male adult, 1do F = 1-day-old female adult, 4do M = 4-day-old male adult, 4do F = 4-day-old female adult. Statistical differences are denoted with different letters, as determined by a one-way ANOVA with Tukey's multiple comparison on log-transformed data ($p < 0.05$). Data represents the mean ± SEM (n = 9).

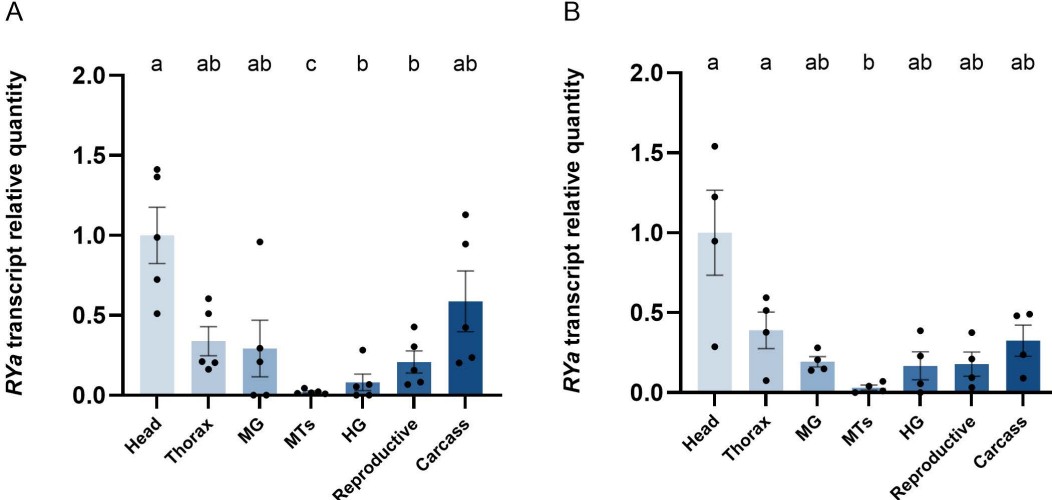

**Fig 2. Organ-specific expression profile of RYamide (RYa) transcript in male (A) and female (B) 1-day-old adult *Ae. aegypti* determined by RT-qPCR.** Transcript levels of various tissues/organs relative to the head. Tissue/organ abbreviations: head, thorax, midgut (MG), Malpighian tubules (MTs), hindgut (HG), reproductive system and carcass. Statistical differences are denoted with different letters, as determined by a one-way ANOVA with Tukey's multiple comparison on log-transformed data ($p < 0.05$). Data represents the mean ± SEM (n = 4-5).

A variety of mosquito neuropeptides that share the RF/RYamide sequence at the C-terminus (Supplementary S2 Table) were tested to examine the sensitivity and specificity of the two putative RYaRs. At a high dosage ($10^{-6}$M), which is considered a saturating concentration [19,57,68], *Aedae*RYa1 and *Aedae*RYa2 elicited strong luminescent responses in RYaR1, which were significantly stronger than the BSA assay media alone, as well as other tested ligands (Fig 3A). Moreover, certain structurally related peptidergic ligands, including full length sNPF1, sNPF2, FMRFa3, FMRFa9, and NPF, were able to induce low luminescent responses by the RYaR1 that were statistically significant from BSA assay media alone, while RYaR1 was unresponsive to other tested neuropeptides which yielded only background level of luminescence (Fig 3A). In contrast, RYaR2 was less selective at the high saturating concentration ($10^{-6}$M) since all tested neuropeptides were found to activate the receptor with levels of luminescence detected that were all significantly higher compared to the negative control with BSA assay media alone (Fig 3B). It is worth noting that *Aedae*RYa1 still exhibited the strongest response in activating RYaR2 among all tested ligands.

A dose-response analysis of RYaR1 revealed that *Aedae*RYa1 has a slightly higher (~3.5-fold greater) activity in terms of receptor activation with a half maximal effective concentration ($EC_{50}$) of 0.47nM compared to *Aedae*RYa2 with a $EC_{50}$ of 1.64nM (Fig 3C). Conversely, RYaR2 shows no capability to distinguish between the two RYamides (*Aedae*RYa1 $EC_{50}$ = 14.08pM; *Aedae*RYa2 $EC_{50}$ = 12.95pM) (Fig 3D). Moreover, the results suggest that RYaR2 is more sensitive to both RYamides in comparison to RYaR1, since pico-molar concentrations of RYamides are able to activate RYaR2. RYamide induced luminescent response of RYaR2 reached the plateau at a concentration ~$10^{-10}$M, while a much higher saturation concentration was measured in RYaR1 (~$10^{-7}$M) (Fig 3C-3D). On the other hand, RYaR1 was more selective in terms of differentiating between RYamides and other structurally related neuropeptides. Only low luminescent responses were detected when NPF, sNPFs, and FMRFas were applied against RYaR1 at high doses, with $EC_{50}$ values approximately in the millimolar to micromolar range (Fig 3C). Comparatively, RYaR2 is activated by NPF, sNPF, and FMRFa more readily, with responses detectable at ~$10^{-8}$M and beyond. Although RYaR2 responded to all tested peptides in the micromolar range, the $EC_{50}$ of RYamides to activate RYaR2 was several orders of magnitude lower compared to these other peptidergic ligands (Fig 3D). Notably, all tested neuropeptides other than RYamides failed to reach the plateau of the dose

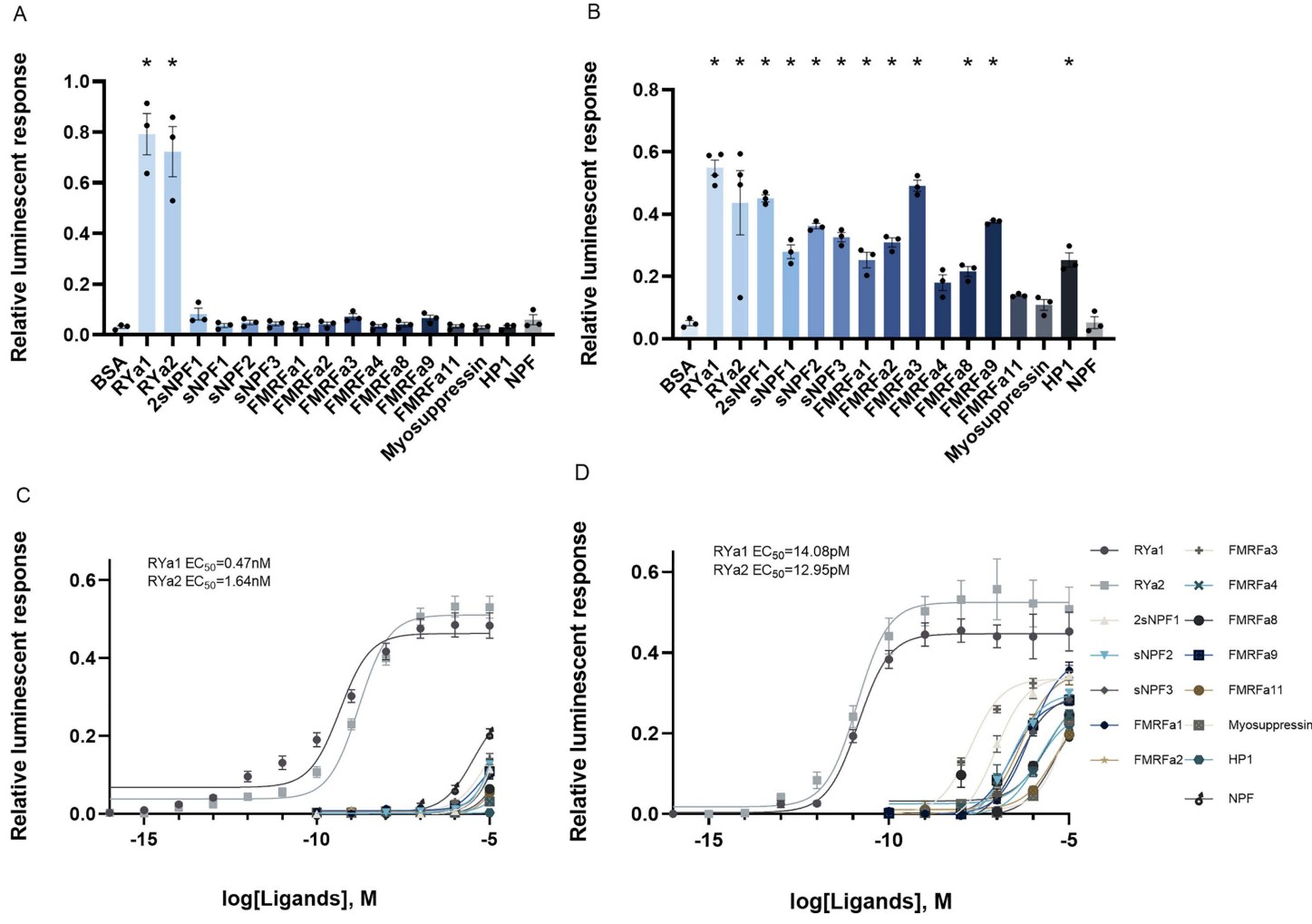

**Fig 3. Luminescent response of CHO-K1 cells expressing *Ae. aegypti* RYaR1 and RYaR2.** RYa1 and RYa2 elicited strong luminescent response activating RYaR1 (A) and RYaR2 (B) at concentration of $10^{-6}$M. Statistical differences are denoted by asterisks (*), as determined by a one-way ANOVA with Dunnett's multiple comparisons test when compared to the BSA (vehicle control) group ($p < 0.05$) (n = 3). Curves were fitted following a sigmoidal dose-response model where RYa1 and RYa2 were found to activate RYaR1 (C) and RYaR2 **(D)**, but with different binding affinities. RYaR1 showed a lower sensitivity compared to RYaR2 with respect to receptor activation by over two orders of magnitude. $EC_{50}$ values shown determined based on averages of all biological replicates (n = 3-6).

response curve (representing maximal receptor activation) at the highest concentration tested ($10^{-5}$M), indicating that RYamides are indeed the natural ligands of these two previously orphan receptors (Fig 3C-3D).

### 3.3 Distribution of RYamide immunoreactivity

To localize RYamide within the central and peripheral nervous system, whole-mount immunohistochemistry was conducted on several tissues of 1-day-old and 4-day-old adult mosquitoes of both sexes using a customized anti-*Aedae*RYa1 antibody (see Methods section). While no immunoreactivity was found in cells localized to the midgut, a ring of prominent RYamide immunoreactivity was observed encircling the pyloric valve, which is localized at the junction between the midgut and hindgut (Fig 4A, 4G, 4H). Strong RYamide immunoreactive staining was observed over the rectum in processes extending into, and in close proximity to the surface of, each of the rectal papillae (pads), with six in females and four

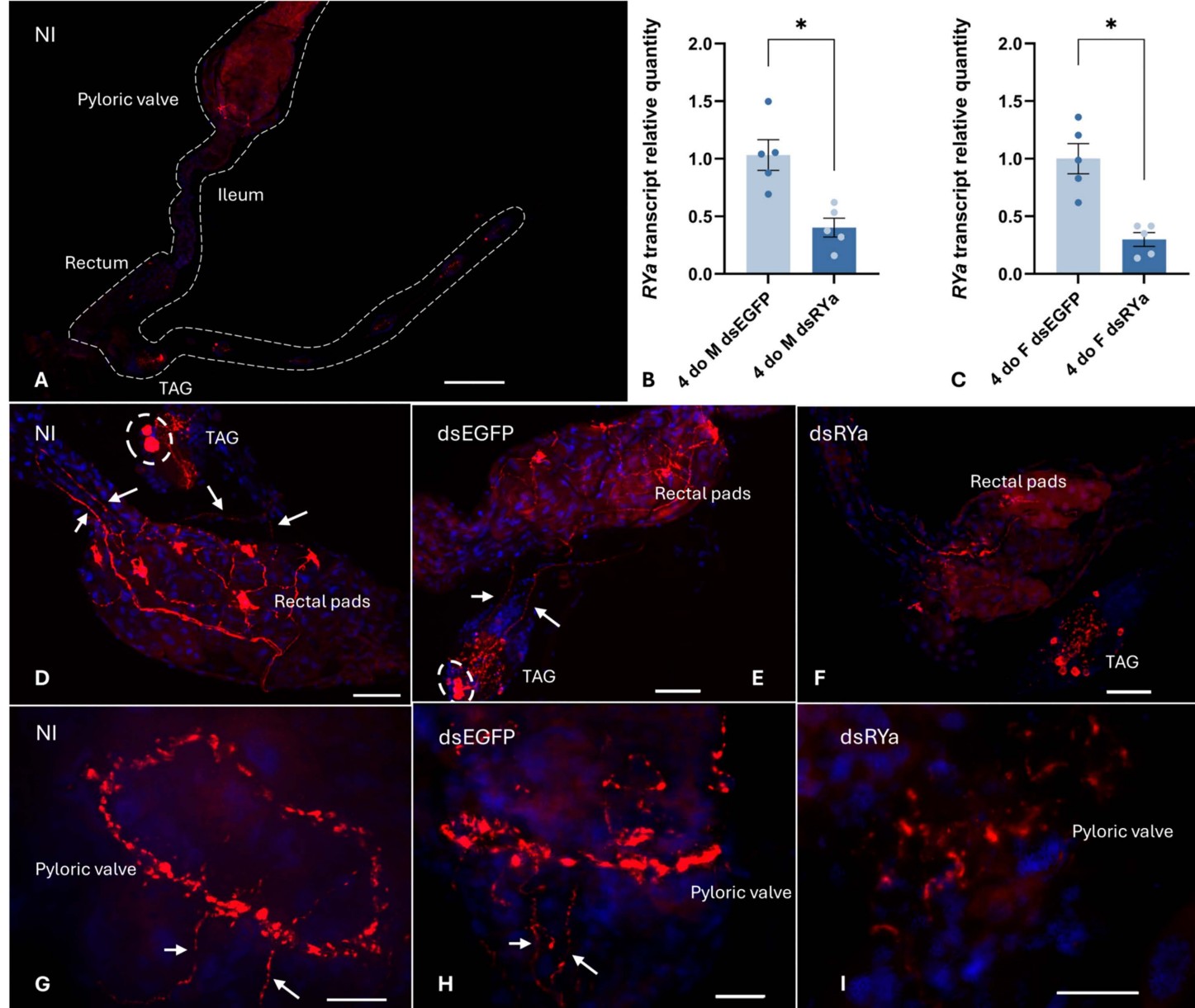

**Fig 4. RYamide immunoreactive cells were localized in the terminal abdominal ganglion with processes extending to the hindgut of 4-day-old adult *Ae. aegypti* reared under standard laboratory conditions via a custom anti-RYa1 antibody.** (A) An overview of RYamide immunoreactivity throughout the gut and abdominal ganglia. The entire gut and the abdominal ganglion are outlined with a dashed line. **(B, C)** Male and female mosquitoes were injected with dsRNAs targeting either EGFP (dsEGFP) or the *Ae. aegypti* RYamide gene (dsRYa). Knockdown efficiency was assessed by RT-qPCR analysis on 4-day-old mosquitoes (3 days post dsRNA injection). Statistical differences are denoted by asterisks (*), as determined by a two-tailed t-test on log-transformed data (p < 0.05) with data represent the mean ± SEM (n = 5). Terminal abdominal ganglion (TAG) and the hindgut of (D) non-injected (NI), (E) dsEGFP-, and (F) dsRYa-injected mosquitoes. Two RYamide producing neurons (encircled by dashed line) were localized in the terminal abdominal ganglion of non-injected and dsEGFP-injected mosquitoes but were not observable in dsRYa-injected mosquitoes. RYamide processes (white arrows) projecting from RYamide-immunoreactive anteromedial neurons of the TAG into the rectal papillae and extending anteriorly over the ileum towards the pyloric valve were observed in non-injected and dsEGFP mosquitoes, while much weaker immunoreactivity was detected in knockdown mosquitoes. (G-I) A ring-shaped RYamide immunoreactive plexus was observed encircling the pyloric valve in non-injected and dsEGFP-injected mosquitoes, while reduced immunostaining intensity was observed in dsRYa-injected mosquitoes. The same pattern of immunolocalization was observed in both males and female adult mosquitoes, with no differences observed between sexes. Scale bars: (A) 200µm; (D-F) 50µm; (G-I) 20µm.

in males (Fig 4A, 4D, 4E). RYamide immunoreactivity was also observed in the CNS, including the ventral nerve cord and the brain. Specifically, a pair of immunoreactive neurons located at the middle of the anterior region of the terminal abdominal ganglion (TAG) were detected. Additionally, RYamide immunoreactive axonal projections emanating from this pair of neurons extended over the rectum and, as mentioned above, associated with each of the rectal papillae (Fig 4D, 4E). Furthermore, this same pair of RYamide immunoreactive processes projecting from the terminal abdominal ganglion to the rectum also continued anteriorly over the ileum towards the pyloric valve (Fig 4D, 4E, 4G, 4H). A few scattered neurons on either side of the terminal ganglion with much weaker immunoreactive staining were also detected, which as described below, may reflect non-specific staining with the antibody (Fig 4D). Multiple immunoreactive neurons detected using the anti-RYamide antibody were bilaterally localized in the lateral protocerebrum and mushroom bodies of the brain (Fig 5). Moreover, RYamide immunoreactivity was found across the bilaterally paired optic lobes, and specifically clustered in the lamina regions of the brain.

To clarify possible cross-reactivity between the anti-RYamide antibody with neuropeptides that share similar structural features to *Ae. aegypti* RYamides, as well as to confirm the localization of RYamide, whole-mount immunohistochemistry was also conducted following *RYamide* knockdown. The efficiency of *RYamide* knockdown was determined by RT-qPCR indicating approximately 60% and 70% reduction in *RYamide* transcript abundance in male and female mosquitoes, respectively (Fig 4B, 4C). In comparison to non-injected and dsEGFP-injected control mosquitoes, intensive RYamide immunoreactivity within each of the rectal papillae was no longer observable in *RYamide* knockdown mosquitoes, whereas only faintly stained RYamide immunoreactive processes were visible over the ileum and rectum (Fig 4D-4F). Similarly, RYamide immunoreactive staining in processes over the pyloric valve was greatly reduced in dsRYa-injected mosquitoes (Fig 4G-4I). Furthermore, while the strongly staining pair of RYamide immunoreactive neurons localized in the middle of the anterior region of the terminal abdominal ganglion were no longer detectable, other immunoreactive neurons localized on either side of the terminal abdominal ganglion were still observable in RYamide knockdown mosquitoes (Fig 4D-4F). This supports the notion that the strongly stained RYamide immunoreactive pair of neurons in the anterior of the terminal abdominal ganglion is the source of RYamide, but other immunoreactive cells detected in this ganglion are likely not RYamide producing neurons. In the brain, all RYamide immunoreactive neurons localized in the protocerebrum and mushroom body as well as most interneuron staining across the optic lobes and lamina regions (Fig 5A-5B) were no longer visible following *RYamide* knockdown (Fig 5C).

### 3.4 Phylogenetic analysis of RYamide receptors among insects and other species

While earlier studies on *D. melanogaster* and *T. castaneum* have only characterized and functionally deorphanized a single RYamide receptor, the fact that *Ae. aegypti* possesses two *bona fide* RYamide receptors encoded by two separate genes is fascinating. This observation raises the question of whether the presence of two RYamide receptors is unique to *Ae. aegypti* mosquitoes or a more common phenomenon among other arthropods. Phylogenetic analysis using maximum-likelihood methods was performed to investigate the evolutionary relationships of RYamide receptors across arthropods and other animals and to gain insights into their evolutionary history. The concise phylogenetic tree (Fig 6) and a more comprehensive tree (Supplementary S4 Fig) including RYamide/luqin-type receptors from more diverse and a greater number of taxa revealed that *Ae. aegypti* RYaR1 (AAEL017005) and RyaR2 (AAEL019786) grouped into two separate clades, positioned as sister groups to each other. This sequence relationship supports the presence of two distinct RYamide receptors in *Ae. aegypti* and corroborates the functional deorphanization findings that revealed these two receptors are highly selective for endogenous RYamide peptides. Even more intriguing is that only the Culicidae family possesses two types of RYamide receptors. Specifically, species in the Culicinae subfamily, such as *Toxorhynchites rutilus septentrionalis*, *Malaya genurostris*, *Wyeomyia smithii*, *Culex pipiens pallens*, *Aedes albopictus*, and *Aedes aegypti*, have both RyaR1 and RyaR2 receptors, while members of the *Anophelinae* subfamily (e.g., *Anopheles gambiae* and *Anopheles coluzzii*) possess only a single receptor, namely RyaR1. Other species within the order Diptera also possess only

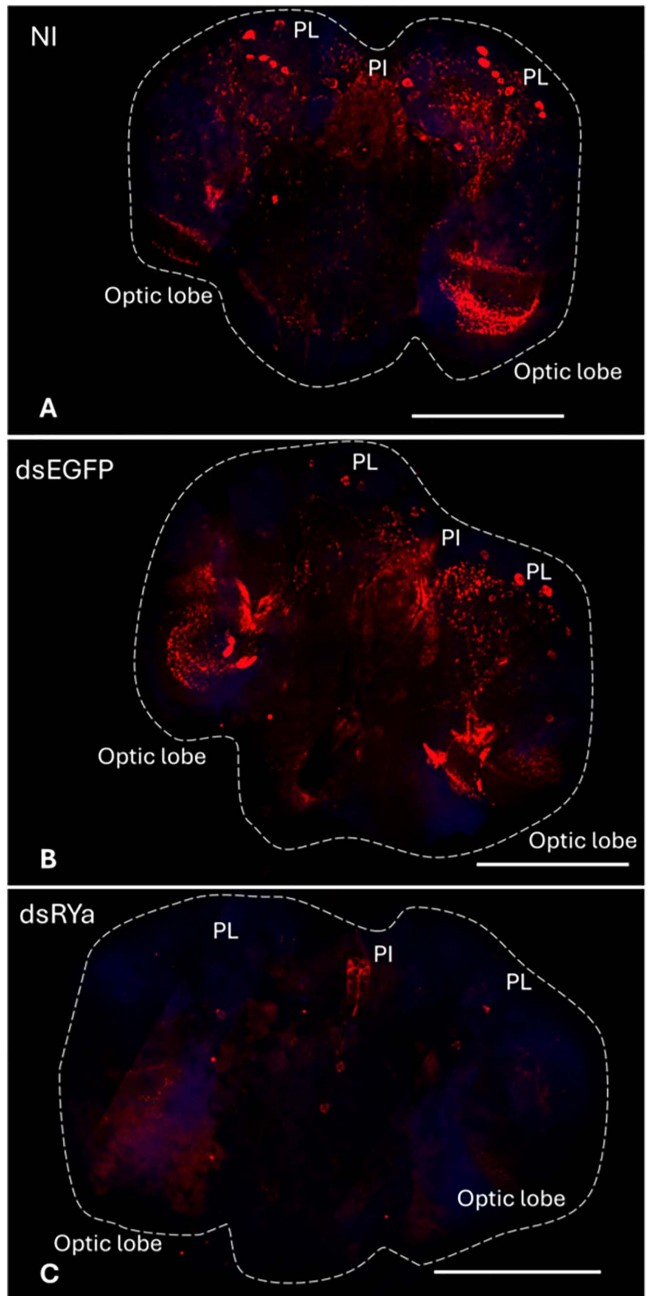

**Fig 5. RYamide immunoreactivity in the brain of 4-day-old adult *Ae. aegypti* reared under standard laboratory conditions via custom anti-*Aedae*RYa1 antibody.** RYamide-like immunoreactivity was localized in several pairs of neurons within the anterior lateral protocerebrum, the *pars latera-lis* (PL), while immunoreactive processes were observed in the medulla of the optic lobe. In each panel, the entire brain is outlined with a dashed line. PI: *pars intercerebralis*. **(A)** Non-injected (control) mosquitoes. (B) dsEGFP mosquitoes. (C) dsRYa mosquitoes. In general, the same pattern and distribution of RYamide-immunoreactivity was observed in the brain of male and female mosquitoes. Scale bars: 200µm.

RyaR1 and are positioned as a sister group to the mosquito RyaR1 clade. Aside from dipterans, representative species from the orders Coleoptera, Hemiptera, Blattodea and Hymenoptera also have receptors that share phylogenetic relationship with dipteran RyaR1. On the other hand, receptors from representative species of Lepidoptera (butterflies and moths)

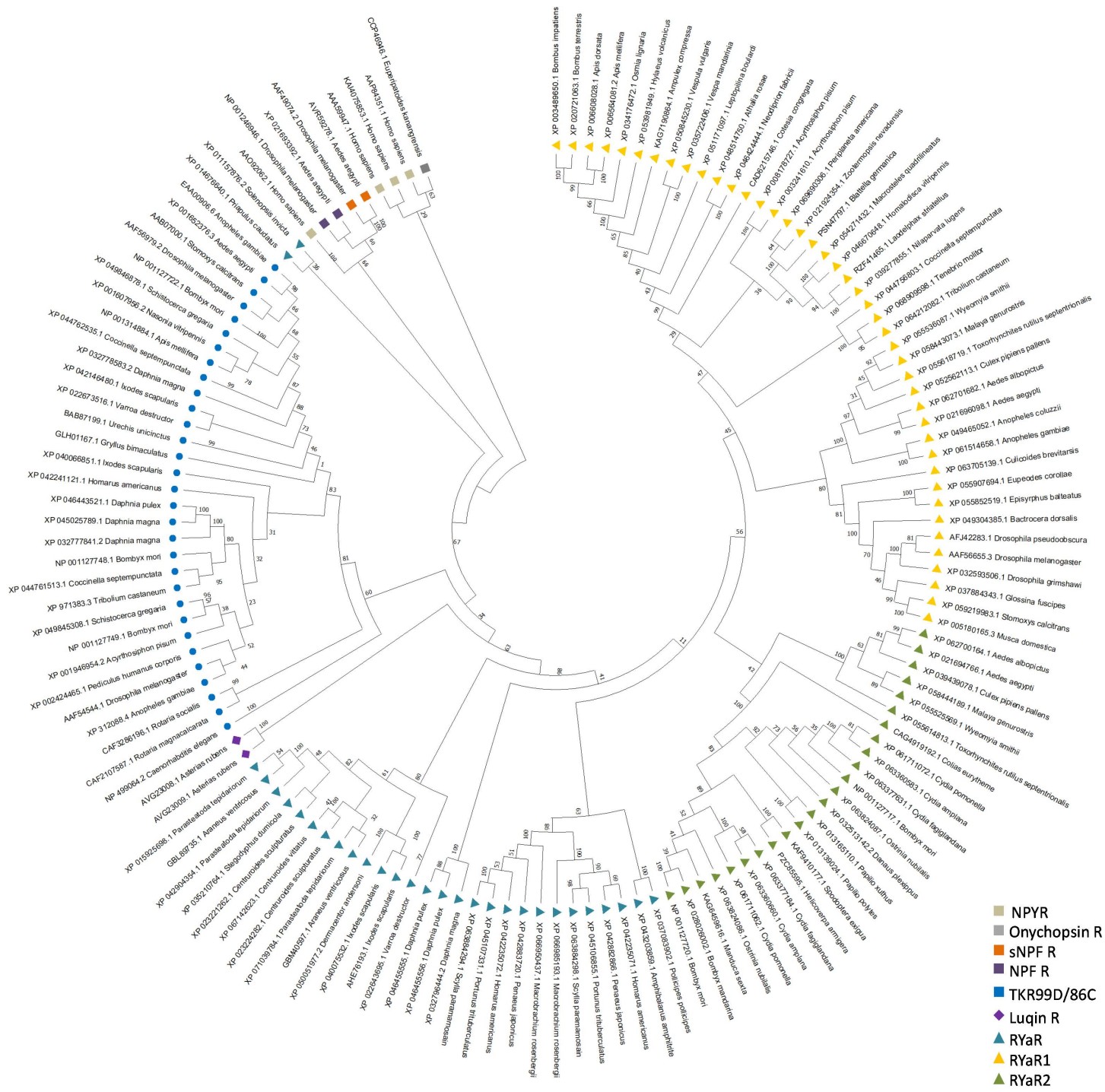

**Fig 6. Phylogenetic analysis showing the relationship of RYamide-related receptors across arthropods and more widely in more diverse organisms.** The tree was constructed using maximum-likelihood phylogenetic analysis methods (with 1000 bootstrap replicates). The numbers adjacent to nodes indicate the statistical support for the clustering of related sequences within the respective clade. Various tachykinin, NPF and sNPF receptors, as well as NPY receptors of *Homo sapiens*, were included to demonstrate the evolutionary relationship between vertebrate NPY receptors and invertebrate RYamide, NPF, and sNPF receptors. *Homo sapiens* NPYR receptors was included in the analysis and imposed as the outgroup.

were positioned within a sister group to the mosquito Culicinae subfamily RyaR2 monophyletic clade (Fig 6). Additionally, evidence of a potential RyaR2 gene duplication was observed in moths, such as *Bombyx*, *Cydia* and *Ostrinia*, but no evidence supporting a RyaR2 gene duplication was observed in butterflies. Notably, an RYamide receptor ortholog of either subtype was not found in available orthopteran (grasshoppers, crickets and locusts) online datasets or in the long-established model for insect physiology, *Rhodnius prolixus*, suggesting selective loss of this system broadly in specific orders (e.g., Orthoptera) and more specifically in a subset of species in other orders (e.g., Hemiptera). Beyond the class Insecta, phylogenetic analysis revealed RYamide-related receptors in other arthropods, including in decapod (crabs, lobsters, crayfish, shrimp, and prawns), thecostracan (barnacles) and branchiopod (water fleas) crustaceans with evidence of expansion in decapods (Fig 6). In another ecdysozoan phylum, Nematoda, the analysis supported the occurrence of RYamide receptor orthologs in several species (Supplementary S4 Fig). Furthermore, RYamide-like receptors were also detected in a variety of arachnids including representatives from the superorder Parasitiformes (ticks and mites), and the orders Araneae (spiders) and Scorpiones (scorpions), which clustered as separate groups basal to the monophyletic group consisting of crustacean and insect RYamide receptors. Our expanded analysis provides evidence supporting the occurrence of RYamide/luqin-related receptors in diverse molluscan, annelid and priapulida representative taxa (Supplementary S4 Fig).

## 4. Discussion

The RYamide neuropeptides and their associated GPCRs were identified across arthropods over a decade ago [18,19,57]. However, research on RYamide distribution, their associated receptors and their phylogeny, and the physiological roles of RYamide signaling in many other insects, including *Ae. aegypti*, is largely unknown. Studying RYamides and their receptors advances our understanding of neuropeptide signaling, which can be conserved or divergent across species and may provide insight for novel pest control strategies if such pathways regulate critical biological functions and could be disrupted using environmentally benign agents.

### 4.1 Deorphanization and evolutionary insights of RYamide receptors

Our findings represent a breakthrough in the deorphanization of GPCRs encoded by the genes AAEL017005 and AAEL019786, validating them as authentic RYamide receptors in *Ae. aegypti*. No study had previously linked the orphan receptors NPYLR5 and NPYLR7 with their RYamide peptide ligands [13,46]. Herein, we provide evidence demonstrating that *Ae. aegypti* NPYLR5 and GPCR-AAEL017005 as well as NPYLR7 and GPCR-AAEL019786, are identical with regards to their deduced amino acid sequences and, as a result of identifying their endogenous and highly active RYamide ligands, accordingly we renamed them as *Aedae*RYaR1 and *Aedae*RYaR2, respectively.

   *Aedae*RYaR1 exhibited a nearly equal response to its two endogenous RYamides peptide ligands, which aligns with relative ligand potencies reported earlier for RYaR in *D. melanogaster* [19,57]. However, the binding and activation of RYaR1 by RYamides in *Ae. aegypti* is more sensitive in terms of the half-maximal response ($EC_{50}$), which is lower compared to receptor orthologs in *T. castaneum* [57], *Caenorhabditis elegans* [69], and *B. mori* [39]. Furthermore, *Ae. aegypti* RYaR2 demonstrated a more potent binding and activation sensitivity to RYamides with a pico-molar range $EC_{50}$, which is orders of magnitude more sensitive compared to *Ae. aegypti* RYaR1 or any previously described RYamide-related receptor [19,39,57,69]. However, while it is unlikely to hold any physiological relevance, the specificity and selectivity of *Ae. aegypti* RYaR1 and RYaR2 was compromised at higher concentrations (particularly for the latter) as there was some promiscuous activation evident by neuropeptides sharing some structural similarities to RYamides including members of the RFamide peptide family. This cross activation of *Ae. aegypti* RYamide receptors by other structurally related ligands was not totally unexpected, as it has been observed in functionally deorphanized orthologous receptors in other arthropods [19,39,57,69].

   To determine whether the presence of a second RYamide receptor is unique to *Ae. aegypti* and other mosquitoes, or if it is more widely found across arthropods, a phylogenetic analysis was conducted to examine the evolutionary relationship

between two RYamide receptor sequences across various arthropod orders. The phylogenetic analysis revealed that only one of the two currently recognized mosquito subfamilies [70–72], specifically members of the Culicinae, possess two RYamide receptor subtypes. Astonishingly, members of the Anophelinae subfamily including *Anopheles* mosquitoes, along with other dipteran species such as *Drosophila* and *Musca domestica*, have only a single RYamide receptor, RYaR1. Relatedly, our analysis also revealed that diverse members of the orders Hymenoptera, Blattodea, Coleoptera Siphonaptera and Hemiptera orders also have a receptor isoform sharing closest similarity to the RYaR1 subtype. Comparatively, Lepidoptera possess only RYaR2 with evidence of duplication of this receptor subtype within this order. Interestingly, we did not find any RYamide-related receptors in genomic data from members of the orders Orthoptera and Trichoptera, while RYamide receptor orthologs were identified in available genomes from species in the orders Odonata, Thysanoptera and Psocodea. Beyond the class Insecta, the phylogenetic analysis indicates that other arthropods possess RYamide-related receptors including members of the subphylum Crustacea such as barnacles and decapod crustaceans, with evidence of expansion in the latter. In Chelicerta, RYamide-related receptors are present in scorpions, spiders and ticks, with evidence of expansion in all but the latter group. Notably, we did not find any evidence of RYamide-related receptors in available genome data from members of the subphylum Myriapoda (millipedes and centipedes). In other protostomes, RYamide-related receptors were identified in Mollusca with presence in cephalopods (octopus), bivalves (mussel, oyster) and gastropods (limpet) where evidence supports expansion in the latter two molluscan classes (bivalves and gastropods). Our analysis also detected RYamide-related receptors within members of the Nematoda, Nemertea, Tardigrada, Brachiopoda, Annelida and Priapulida phyla. The current data supports that an ancestral RYamide receptor in arthropods along with other protostomes and the Luqin-type receptors in Ambulacraria [73], such as luqin receptor 1 and 2 characterized in the starfish *Asterias rubens* [17], share a common ancestor. Indeed, luqin-type and, relatedly, RYamide-related signaling originated in a common ancestor of bilaterian organisms and is paralog of tachykinin-like signaling systems, while these two clades are evolutionarily related to NPY/NPF related signaling [17,73]. Based on the data in insects, the results suggest a gene duplication of RYa1R occurred in a Mecopterida ancestor prior to the divergence of Antliophora and Amphiesmenoptera superorders within the Holometabola [74,75], resulting in a second receptor subtype (RYaR2) retained in Lepidoptera and both receptor subtypes occurring in select dipterans, specifically in the Culicinae subfamily of mosquitoes. Collectively, RYamide receptors are broadly distributed in arthropods and are orthologs of luqin-type receptors functionally characterized in deuterostomes [17] supporting earlier findings that RYamide/luqin-type receptors and their ligands are evolutionarily ancient signaling systems found across bilaterian organisms [16], despite being absent in chordates [73].

## 4.2 The nervous system as source and the hindgut as target

The expression profile of *RYamide* transcript was examined during developmental and in individual organs/tissues of early and later stage adult *Ae. aegypti*. The developmental expression profile revealed enrichment of *RYamide* transcripts following the transition from pupal to adult stages. Specifically, the enrichment of *RYamide* was detected in both early and later stage male adults relative to fourth instar larvae, and this could be indicative of a sex-specific post-eclosion function for RYamide signaling. In contrast to these findings, the number of RYamide expressing neurons in the nervous system including the brain and the abdominal ganglia, as well as in enteroendocrine cells in the midgut of *B. mori*, reduced over development, as the highest number of RYamide expressing cells was visualized in larva of *B. mori* [21].

Spatial transcript expression profiling and immunolocalization of RYamide in adult *Ae. aegypti* aimed to unveil potential physiological functions for RYamide and its corresponding receptors by determining RYamide producing cells and target tissues/organs. *RYamide* transcript was found to be highly enriched in the brain of both sexes, which aligns with publicly available RNA-seq data [29] that indicates enrichment in the head of female *Ae. aegypti* (Supplementary S1 Fig). This also corroborates RYamide producing neurons immunolocalized in the brain and confirmed through RNAi-mediated knockdown. In particular, clusters of RYamide immunoreactive neurons were localized in the anterior lateral protocerebrum region of the brain along with the area above the calyx region covered by intrinsic neurons, Kenyon cell bodies [76]

within the mushroom body. Insect mushroom bodies are complex in their neuronal connections and, unsurprisingly, are known to be involved in diverse functions such as learning and memory [77–80], sensory information processing, including olfactory [81], acoustic [82,83], and gustatory information [84,85], and sleep regulation [86]. Interestingly, extrinsic neurons, such as octopaminergic and dopaminergic neurons, which form synapses on mushroom body Kenyon cell axonal projections, have been suggested to be associated with feeding-related behavioural responses [87,88]. The previously reported feeding suppression role of RYamide (Maeda et al., 2015) may be linked to Kenyon cells in the mushroom body, given the immunolocalization of RYamide and the high enrichment of RYaR1 in the brain [29] (see Supplementary S3 Fig. A). Taken together, this suggests that the brain is one of the main sources of RYamide in mosquitoes. Furthermore, RYamide has been reported near the anterior edge of the brain in mated, previtellogenic female *Ae. aegypti* [89], supporting our findings. Our study strengthens the evidence for the RYamide localization through RNAi-mediated knockdown, offering a more detailed view of the specific clusters of RYamide-producing neurons and confirming that both sexes exhibit the same expression pattern. Furthermore, the results are generally consistent with RYamide expression reported in the adult *B. mori* brain, where RYamide-expressing cells include a pair of protocerebral posterior dorsomedial (PDM) and protocerebral anterolateral (PAL) neurons [21]. Unlike the two RYamide expressing posterior ventrolateral (PVL) neurons observed at the base of optic lobes in *B. mori*, strong RYamide immunoreactivity in processes was detected in the medulla of the optic lobes in adult mosquitoes.

In addition to the brain, *RYamide* transcript was enriched in the carcass, comprised of the fat body, epidermis, cuticle, and abdominal ganglia. Notably, a pair of neurons in the anteromedial region of the terminal abdominal ganglion exhibited intense RYamide immunoreactivity (Fig 4D-4F). These findings indicate that the ventral nerve cord, a key part of the mosquito nervous system, is also a source of RYamide. In support of this observation, the presence of RYa1 in the terminal abdominal ganglion in *Ae. aegypti* was confirmed by an earlier peptidomic study through mass spectrometry [18]. On the other hand, the two RYamide-expressing cells detected in the larval and pupal stages of *B. mori* were no longer observed in the adult [21], implying that RYamide produced in the terminal abdominal ganglia of adult mosquitoes may serve a unique role. The terminal abdominal ganglion in mosquitoes is a neural structure that regulates reproductive [90] and hindgut-related functions [38]. For example, surgically removing the terminal abdominal ganglion of a post-blood meal female mosquito results in the accumulation of waste products from digested blood, including uric acid and hematin [91], in the gut lumen [38], while also significantly reduced water loss.

In addition to the detection of RYamide immunoreactive neurosecretory cells in the terminal abdominal ganglion of adult mosquitoes, prominent axonal projections originating from these neurons were observed extending towards the rectal pads (or rectal papillae) within the rectum along with processes extending over length of the ileum and terminating over the pyloric valve forming a circular plexus. The rectal pads are the primary site responsible for actively reabsorbing water and ions from the gut lumen to the haemolymph, helping to achieve a key function of the hindgut in excreting nitrogenous waste [33,35,36]. Previous studies noted that the ultrastructure of rectal papillae in 0–6-hour post-blood meal female mosquitoes exhibits more elaborate infolding on the apical membrane, suggesting increased activity within the rectal papillae during the peak diuresis period [45,92]. In both larval and adult mosquito rectal papillae, P-type $Na^+/K^+$-ATPase and V-type $H^+$-ATPase have been immunolocalized on the basal and apical surfaces, respectively [42]. Reabsorption of $K^+$ is mediated by V-type $H^+$-ATPase, while $Na^+$ reabsorption is coupled to the apical $Na^+/H^+$ exchangers and basolateral P-type $Na^+/K^+$ ATPases [93]. Interestingly, available RNA-seq data indicates that RYaR2 is highly enriched in the hindgut of sucrose-fed female mosquitoes [29] (see Supplementary S3B in S3 Fig). In support of this insight from the public database, a recent preprint study demonstrated that RYaR2 (referred therein as NPYLR7) is expressed exclusively in rectal pad cells of the hindgut, which were found to elicit calcium signaling in response to RYamide peptide [94]. Interestingly, deletion of RYaR2 disrupts egg viability and protein provisioning after a blood meal, but it did not impair overall fluid regulation following blood-feeding, a simulated isotonic saline meal or sucrose feeding by female *Ae. aegypti* [94]. In line with these recent findings, RYaR2 activation has been previously shown to induce suppression of host-seeking behaviour [13]. However,

considering the enrichment of RYamide signaling in males as well as females, including the prominent immunoreactive staining associated with the hindgut, the current findings provide evidence that RYamide may participate in excretory functions at times not requiring a pronounced diuresis.

In conclusion, this study expands our understanding of RYamide and its receptors in *Ae. aegypti*, one of the most successful human disease vectors. By characterizing the developmental and tissue-specific expression of the *RYamide* transcript, along with the distribution of RYamide immunoreactivity, we demonstrate that the brain and ventral nerve cord serve as the primary sites of RYamide production, while the rectum appears as a key target of RYamide and RYaR2 signaling. The close association between RYamide immunoreactive processes and the rectal pads in adult male and female mosquitoes suggests a potential role in ionoregulatory processes, presenting a promising direction for further investigation. Currently, only a limited number of RYamide receptors have been deorphanized in arthropods, and we functionally characterized two GPCRs in *Ae. aegypti*, RYaR1 (AAEL017005) and RYaR2 (AAEL019786), as *bona fide* RYamide receptors. Both receptors exhibit high specificity for the endogenous *Ae. aegypti* RYamides, while other tested neuropeptides with a similar RF/Yamide C-terminal motif were several orders of magnitude less effective in activating them. NPY-related signaling pathways are closely associated with foraging and feeding-related behaviors across both invertebrates and vertebrates, including insects, mice, and humans [95–97]. Reflecting this functional conservation, the sequences of vertebrate NPYRs and invertebrate NPYLRs exhibit a high degree of similarity. It was previously reported that invertebrate NPYLRs share approximately 60% sequence similarity with vertebrate NPY Y2 receptors [22]. Interestingly, NPYLR7 (now known as *Ae. aegypti*, RYaR2) has been shown to suppress host-seeking behavior in female mosquitoes upon activation with small molecule agonists [13]. Herein, NPYLR7 has been deorphanized and renamed as RYaR2, which provides new insights and directions for the RYamide story. First, our findings indicate that RYamide and RYaR2 signaling are strongly associated with the rectal pads, with RYaR2 shown recently to be localized to a non-neuronal population of cells [94] within the rectal pads, suggesting a potential role in iono- and/or osmoregulation activity by the rectum of adult mosquitoes. This raises intriguing questions about whether the suppression of host-seeking behavior and osmoregulatory functions are interconnected. Additionally, it remains unclear whether NPYLR7/ RYaR2 mediates distinct physiological roles in male mosquitoes given the enrichment of RYamide transcript in males and the strong immunoreactivity associated with the hindgut originating from RYamide producing neurons within the terminal abdominal ganglion, matching the distribution observed in female mosquitoes. Secondly, small-molecule agonists targeting NPYLR7/ RYaR2 have been developed that suppress host-seeking behavior in mosquitoes. Now that RYamide has been identified as the endogenous ligand of NPYLR7/ RYaR2, this offers valuable insights for the structure-guided design of novel agonists with greater potency and selectivity. Given the importance NPYLR7/ RYaR2 in driving the suppression of host-seeking, biting and attraction to human hosts [13,89,94], further studies on this signaling pathway could yield innovative control strategies aimed at preventing mosquito biting behavior and, as a result, mitigate disease transmission.

## Supporting information

**S1 Fig. Organ-specific expression profile of RYamide in adult *Ae. aegypti* based on public RNAseq data.** Figure was prepared using available RNA-seq dataset (Hixson *et al.* 2022).
(TIF)

**S2 Fig. Normalized luminescent response of CHO-K1 cells expressing EGFP in pBud vector, *Ae. aegypti* RYaR2 (AAEL019786) in pcDNA3.1 + vector (n = 3).** Assay media (BSA) alone and different ligands ($10^{-6}$M) were applied to cells expressing different expression constructs to validate the receptor activity by comparing the luminescent responses generated via receptor activation. Statistical differences are denoted with asterisk (*), as determined by a two-way ANOVA with Šídák's multiple comparisons test ($p < 0.05$).
(TIFF)

**S3 Fig. Organ-specific expression profile of RYaR1 (A) and RYaR2 (B) in adult *Ae. aegypti* based on public RNA-seq data.** Figure was prepared using available RNA-seq dataset (Hixson *et al.* 2022).
(TIF)

**S4 Fig. Phylogenetic relationship of RYamide receptors across protostomes and deuterostomes.** Tree was constructed using maximum-likelihood phylogenetic analysis methods (with 1000 bootstrap replicates). The annotated numbers adjacent to nodes indicate the support percentage for the clustering of related sequences within the respective clade. Tachykinin, NPF and sNPF receptors, as well as NPY receptors of *Homo sapiens*, were included to demonstrate the evolutionary relationship between vertebrate NPY receptors and invertebrate RYamide, tachykinin, NPF, and sNPF receptors. *Homo sapiens* NPYR was included in the analysis and imposed as the outgroup.
(PDF)

**S1 Table. Information of designed primers.** Geneious Pro Bioinformatics Software was used to design primers.
(DOCX)

**S2 Table. Name and primary amino acid sequences of *Ae. aegypti* neuropeptides used in immunohistochemistry and functional bioluminescence assay.** Highlighted sequences indicate the conservation of residues at the C-terminus.
(DOCX)

## Author contributions

**Conceptualization:** Jinghan Tan, Thomas Luong, Jean-Paul V. Paluzzi.

**Data curation:** Jinghan Tan, Thomas Luong.

**Formal analysis:** Jinghan Tan, Thomas Luong, Jean-Paul V. Paluzzi.

**Funding acquisition:** Jean-Paul V. Paluzzi.

**Investigation:** Jinghan Tan, Thomas Luong.

**Methodology:** Jinghan Tan, Thomas Luong.

**Project administration:** Jean-Paul V. Paluzzi.

**Resources:** Jean-Paul V. Paluzzi.

**Software:** Jean-Paul V. Paluzzi.

**Supervision:** Jean-Paul V. Paluzzi.

**Validation:** Jinghan Tan, Jean-Paul V. Paluzzi.

**Visualization:** Jinghan Tan, Jean-Paul V. Paluzzi.

**Writing – original draft:** Jinghan Tan.

**Writing – review & editing:** Jinghan Tan, Jean-Paul V. Paluzzi.

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
