## [Decision Letter · Decision Letter 0]

30 Nov 2025

Dear Dr. Paluzzi,

Thank you for submitting your manuscript to PLOS ONE. After careful consideration, we feel that it has merit but does not fully meet PLOS ONE’s publication criteria as it currently stands. Therefore, we invite you to submit a revised version of the manuscript that addresses the points raised during the review process.

We look forward to receiving your revised manuscript.

Kind regards,

Pedro L. Oliveira

Academic Editor

PLOS ONE

Journal Requirements:

This research was funded by the Natural Sciences and Engineering Research Council of Canada (NSERC) Discovery Grant (JPP), Ontario Ministry of Research Innovation Early Researcher Award (JPP)

4. Please amend your authorship list in your manuscript file to include author Jean-Paul V. Paluzzi, Jinghan Tan, Thomas Luong.

5. Please amend the manuscript submission data (via Edit Submission) to include author Paluzzi, JP, Tan, J, Luong, T.

Reviewers' comments:

Reviewer's Responses to Questions

**Comments to the Author**

1. Is the manuscript technically sound, and do the data support the conclusions?

Reviewer #1: Partly

2. Has the statistical analysis been performed appropriately and rigorously?

Reviewer #1: Yes

3. Have the authors made all data underlying the findings in their manuscript fully available?

Reviewer #1: Yes

4. Is the manuscript presented in an intelligible fashion and written in standard English?

Reviewer #1: Yes

Reviewer #1: The manuscript aims to deorphanize two putative GPCRs from the mosquito Aedes aegypti.

The manuscript is well written, but the introduction could be shortened a little bit. Also in the results section the authors present some information about the literature that should be moved to the discussion (line 438).

The possible link to the hindgut physiology is based on the higher staining for the Ryamide in that region. There is no additional evidence to show that the head is the source and the hindgut a target. I would suggest removing that statement from the title because the only evidence of this biological role is a positive immunostaining. More evidence is needed.

Is it possible to test the expression of the putative receptors in the different tissues, particularly in the hindgut? That would add more evidence to the hypothesis of the head being the source (RYamide staining) and the hindgut a target by having a high expression of the receptors.

The immunofluorescence images are very informative however the contrast is not very good. It is hard to visualize the tissue overall structure. Do the authors have any morphological staining for the general structure of the tissue? If not, it would be good to have an outline of the tissue or a better contrast of the pictures.

Minor comments

- Please add titles to your graphs. For example, in figure 1 – Ryamide expression – Males. Please add to all graphs.

**Do you want your identity to be public for this peer review?** For information about this choice, including consent withdrawal, please see our Privacy Policy

Reviewer #1: No

---

## [Author Response · Author response to Decision Letter 1]

2 Jan 2026

We have appended the detailed response to reviewer comments with our revised manuscript below. However, please see the attached detailed reviewer responses that have our comments in blue coloured text.

Response to reviewer comments/suggestions (in blue font):

Reviewer #1: The manuscript aims to deorphanize two putative GPCRs from the mosquito Aedes aegypti.

The manuscript is well written, but the introduction could be shortened a little bit. Also in the results section the authors present some information about the literature that should be moved to the discussion (line 438).

We thank the reviewer for their positive comments. We have revised the introduction by removing some details from the literature making this section more concise. The last few sentences in the results section have also been revised, as requested.

The possible link to the hindgut physiology is based on the higher staining for the Ryamide in that region. There is no additional evidence to show that the head is the source and the hindgut a target. I would suggest removing that statement from the title because the only evidence of this biological role is a positive immunostaining. More evidence is needed.

We thank the reviewer for this suggestion. As we mentioned in discussion (lines 602–605), based on available RNA-seq datasets (cited in our manuscript and see Supplementary Fig. S3) and a recent preprint study (Greppi et al., 2025), RYaR2 that is now confirmed as an authentic RYamide receptor, is highly enriched in the adult mosquito hindgut. More specifically, RYaR2 is expressed exclusively in rectal pad cells (Greppi et al., 2025), which is an important site of ion transport. These findings strongly support that there is a potential RYamide and RYaR2 signaling pathway associated with the hindgut.

Furthermore, we demonstrate that RYamide is secreted by two neurons in the terminal abdominal ganglia, with axonal projections extending toward the rectal pads. This observation further supports a potential role for RYamide in regulating hindgut-related functions, as the terminal abdominal ganglia have been reported to directly innervate ion transporting regions in the insect hindgut (Cook et al., 1991; Van Handek and Klowden, 1996).

Is it possible to test the expression of the putative receptors in the different tissues, particularly in the hindgut? That would add more evidence to the hypothesis of the head being the source (RYamide staining) and the hindgut a target by having a high expression of the receptors.

We thank the reviewer for this valuable suggestion. The RYaR2 gene is intronless, making it very difficult to accurately assess transcript abundance using qPCR without interference from genomic DNA. We had designed and tried several primer sets but could not reliably quantify transcript expression for this gene (data not shown). Publicly available RNA-seq data indicate that RYaR1 is highly enriched in the head, whereas RYaR2 is expressed almost exclusively in the hindgut of adult mosquitoes (see Supplementary Fig. S3; Hixson et al., 2022). We do have simple RT-PCR (not quantitative) data/results for both receptors but felt the available RNA-seq data (in Fig. S3) is more appropriate.

The immunofluorescence images are very informative however the contrast is not very good. It is hard to visualize the tissue overall structure. Do the authors have any morphological staining for the general structure of the tissue? If not, it would be good to have an outline of the tissue or a better contrast of the pictures.

We thank the reviewer for this feedback. Images have been revised as requested, with the gut, abdominal ganglion, and brain tissues all outlined (specifically in panels with preparations difficult to visualize like Fig. 4a and all panels in Fig. 5).

Minor comments

- Please add titles to your graphs. For example, in figure 1 – Ryamide expression – Males. Please add to all graphs.

We thank the reviewer for this feedback. All figure captions, including figure title sentences, are provided at the end of the manuscript (line 942-1032).

---

## [Editor Report · Decision Letter 1]

21 Jan 2026

Characterization and deorphanization of RYamide signaling in Aedes aegypti: a potential regulator of hindgut-associated physiology

PONE-D-25-55793R1

Dear Dr. Paluzzi,

We’re pleased to inform you that your manuscript has been judged scientifically suitable for publication and will be formally accepted for publication once it meets all outstanding technical requirements.

Kind regards,

Pedro L. Oliveira

Academic Editor

PLOS One
---

## [Editor Report · Acceptance letter]

PONE-D-25-55793R1

PLOS One

Dear Dr. Paluzzi,

I'm pleased to inform you that your manuscript has been deemed suitable for publication in PLOS One. Congratulations! Your manuscript is now being handed over to our production team.

Kind regards,

on behalf of

Dr. Pedro L. Oliveira

Academic Editor

PLOS One